# Ca^2+^-Activated K^+^ Channels in Progenitor Cells of Musculoskeletal Tissues: A Narrative Review

**DOI:** 10.3390/ijms24076796

**Published:** 2023-04-05

**Authors:** Roland Takács, Patrik Kovács, Rana Abdelsattar Ebeid, János Almássy, János Fodor, László Ducza, Richard Barrett-Jolley, Rebecca Lewis, Csaba Matta

**Affiliations:** 1Department of Anatomy, Histology and Embryology, Faculty of Medicine, University of Debrecen, H-4032 Debrecen, Hungary; takacs.roland@med.unideb.hu (R.T.); patrik.kovacs@med.unideb.hu (P.K.); rony.bidoo@gmail.com (R.A.E.); ducza.laszlo@anat.med.unideb.hu (L.D.); 2Department of Physiology, Faculty of Medicine, Semmelweis University, H-1428 Budapest, Hungary; almassy.janos@med.semmelweis-univ.hu; 3Department of Physiology, Faculty of Medicine, University of Debrecen, H-4032 Debrecen, Hungary; fodor.janos@med.unideb.hu; 4Department of Musculoskeletal Biology, Faculty of Health and Life Sciences, Institute of Ageing and Chronic Disease, University of Liverpool, Liverpool L69 3GA, UK; rbj@liverpool.ac.uk; 5Department of Comparative Biomedical Sciences, School of Veterinary Medicine, Faculty of Health and Medical Sciences, University of Surrey, Guildford GU2 7XH, UK; rebecca.lewis@surrey.ac.uk

**Keywords:** chondrogenesis, osteogenesis, muscle differentiation, progenitor cell, osteoarthritis, ion channel, channelome, BK channel, musculoskeletal diseases

## Abstract

Musculoskeletal disorders represent one of the main causes of disability worldwide, and their prevalence is predicted to increase in the coming decades. Stem cell therapy may be a promising option for the treatment of some of the musculoskeletal diseases. Although significant progress has been made in musculoskeletal stem cell research, osteoarthritis, the most-common musculoskeletal disorder, still lacks curative treatment. To fine-tune stem-cell-based therapy, it is necessary to focus on the underlying biological mechanisms. Ion channels and the bioelectric signals they generate control the proliferation, differentiation, and migration of musculoskeletal progenitor cells. Calcium- and voltage-activated potassium (K_Ca_) channels are key players in cell physiology in cells of the musculoskeletal system. This review article focused on the big conductance (BK) K_Ca_ channels. The regulatory function of BK channels requires interactions with diverse sets of proteins that have different functions in tissue-resident stem cells. In this narrative review article, we discuss the main ion channels of musculoskeletal stem cells, with a focus on calcium-dependent potassium channels, especially on the large conductance BK channel. We review their expression and function in progenitor cell proliferation, differentiation, and migration and highlight gaps in current knowledge on their involvement in musculoskeletal diseases.

## 1. Introduction

The musculoskeletal system consists of bones, muscles, tendons, ligaments, and joints—a network that functions together to provide support, movement, and protection for the body.

A healthy musculoskeletal system is essential for maintaining mobility and overall quality of life. Bones provide support for the body and protect vital organs, while muscles allow for movement and help maintain posture. Tendons connect muscles to bones, and ligaments connect bones to other bones, providing stability and preventing excessive movement at the joints [1]. Synovial joints are the most-common type of joint in the human body and are characterised by the presence of a synovial membrane that produces synovial fluid. This fluid helps to reduce friction and wear between the joint surfaces during movement, while also providing nutrients to support the health of the joint structures. Synovial joints allow for a wide range of movements. Examples of synovial joints include the knee, hip, shoulder, and elbow joints [2].

Cartilage, muscle, bone, synovium, tendons, and ligaments are important components of the musculoskeletal system, each with unique features, origins, and functions. Understanding the roles of the progenitor cells that make up these tissues can help promote optimal musculoskeletal health and prevent injuries.

Cartilage is a connective tissue that provides cushioning and support at joints, reducing friction and preventing damage to the underlying bones. Cartilage is composed of chondrocytes, which are derived from mesenchymal stem cells (MSCs). MSCs are multipotent cells that can differentiate into several cell types, including chondrocytes, osteoblasts, and adipocytes [2]. Bone tissue is an essential component of the musculoskeletal system. It is composed of cells called osteoblasts, osteocytes, and osteoclasts, which work together to build and maintain bone tissue. Bones are constantly remodelling, with old bone being broken down by osteoclasts and new bone being formed by osteoblasts [1]. 

Muscle tissues in the musculoskeletal system are composed of specialised cells called muscle fibres, which contract, thereby generating force. Skeletal muscle tissue is derived from myogenic precursor cells that can differentiate into the skeletal muscle fibres attaching to bone, allowing for movement [3].

The synovium is a specialised tissue that lines the inner surface of joints and produces synovial fluid. It is composed of different cell types, including fibroblast-like synoviocytes (FLS), macrophages, and T cells. FLSs are derived from MSCs and are responsible for producing the synovial fluid, as well as maintaining the health and function of the joints [4].

Tendons and ligaments are composed of collagen fibres and are important for maintaining joint stability and allowing for controlled movement. Tenocytes, the cells that produce collagen in tendons and ligaments, are elongated fibroblastic cells, derived from MSCs [3].

### 1.1. Progenitor Cells in Musculoskeletal Tissues

Stem cells are characterised by self-renewal through mitotic cell division and differentiation into tissue-specific cell types. Adult stem cells are an important source for the repair of damaged tissues. MSCs have the potential to form cells of the musculoskeletal system, including osteoblasts, adipocytes, chondrocytes, and myocytes [5]. In the adult musculoskeletal system, tissue-resident MSCs have been described in bone marrow and periosteum, articular cartilage, synovium, muscle, ligament, and tendon [6]. Regenerative medicine aims to exploit the differentiation potential of MSCs to restore damaged musculoskeletal tissues. MSCs have been historically believed to migrate to the sites of tissue injury, where they would engraft, differentiate, and reconstitute the tissue architecture. However, recent data have clearly shown that this is not entirely the case. There is an array of reparative actions of MSCs including enhancing cell viability and proliferation, reducing apoptosis, and local immunomodulation via paracrine mechanisms such as secretion of growth factors and cytokines, cell-to-cell interactions through tunnelling nanotubes mediating the transfer of mitochondria and lysosomal vesicles, and through releasing extracellular vesicles [6]. Induced pluripotent stem cells (iPSCs), derived from a patient’s somatic cells through genetic reprogramming, can generate all tissues in the body. The application of iPSCs also holds the promise in the field of regenerative medicine for clinical orthopaedics [7].

### 1.2. Musculoskeletal Diseases

Maintaining a healthy musculoskeletal system involves a combination of physical activity, proper nutrition, and medical management of any underlying conditions. Exercise, physical activity, and nutrition are essential for building and maintaining healthy bones and muscles, as well as improving flexibility and range of motion [1].

Any discomfort or irreversible or disabling injuries affecting the motor organs including muscles, tendons, bones, cartilage, ligaments, and nerves are classified as musculoskeletal disorders [8]. The five most-common musculoskeletal conditions comprise rheumatoid arthritis (RA), osteoarthritis (OA), low back pain (LBP), neck pain (NP), and gout [8]. Musculoskeletal disorders are common causes of disability; in fact, such disorders are ranked fifth in disability-adjusted life years (DALYs) [9]. While heavy physical work, awkward posture, and high body mass index (BMI) increase the risk of certain musculoskeletal conditions, some of the risk factors of OA are still unknown. The global burden of musculoskeletal disorders is steadily increasing, which highlights a yet unmet need to develop strategies to reverse these trends.

OA is one of the most-common musculoskeletal disorders [10]. One of the hallmarks of OA is the loss of articular cartilage at the joint, and there is currently no cure for this painful condition, only symptomatic treatment. The most-common therapy involves replacement of the degraded tissue, and potentially the surrounding area, with an implant. This has resulted in a substantial improvement in the quality of life of those affected; however, it is only a temporary solution. Prosthetic implants have a relatively short lifespan as implant loosening is inevitable within 10–15 years, despite the significant advances in recent years [11].

Other therapies include encouraging the healing of articular cartilage by removing the damaged tissue, introducing fresh blood supply to the area (microfracture), or injecting chondrocytes into the site (autologous chondrocyte injection) [12]. Osteochondral grafts can also be implanted to replace the damaged cartilage; these can be donated from the patient (autograft) or by a donor (allograft). Although well established, these treatments carry significant risk of complications including pain, joint swelling, and excessive bleeding, depending on the graft location and transplantation method [13,14]. Further solutions include the incorporation of stem cells into the joint, either via articular injection or tissue engineering, whereby stem cells are combined with biodegradable materials [15]. Research aimed at understanding the biological characteristics of MSCs and identifying methodologies by which they potentially contribute to hyaline articular cartilage repair has been going on for decades. MSCs have also been tested in clinical trials for a range of musculoskeletal conditions such as OA, RA, fracture repair, regeneration of articular cartilage, tendon repair, and for the treatment of degenerative disc disease [16]. Despite these efforts, the use of MSCs as a treatment option for these conditions is still uncertain. More details need to be understood concerning the molecular regulation of the differentiation pathways before MSCs can become an effective treatment option for musculoskeletal disorders, including OA.

## 2. Ion Channels Involved in Progenitor Cell Differentiation

Plasma membrane ion channels and transporters are known to play a central role in the proliferation, differentiation, and apoptosis of a wide range of cells, including stem cells. Functional ionic currents have been reported to be heterogeneously present in different types of stem cells [17]. In this narrative review article, after briefly describing the channelome of the progenitor cells of the musculoskeletal system, we summarise current knowledge on the expression and function of a specific class of ion channels, the calcium-dependent potassium (K_Ca_) channels, in health and disease. We highlight gaps in current knowledge regarding K_Ca_ channels in musculoskeletal progenitor cells, which could be exploited and developed into novel therapeutic strategies. We also discuss their pathological implications and their potential as molecular targets for the development of innovative and promising therapeutic strategies in musculoskeletal disorders.

### 2.1. Ion Channels That Mediate Chondrogenesis 

While mature chondrocytes have been extensively studied in terms of the functional expression of channels and transporters including (but not limited to) aquaporin water channels, calcium-activated potassium channels, chloride channels, sodium channels, potassium channels, *N*-methyl D-aspartate (NMDA) channels, transient receptor potential (TRP) channels, and calcium channels [18,19,20], current knowledge is much more limited regarding the channelome of chondroprogenitor cells (CPCs).

During chondrogenic differentiation, undifferentiated progenitor cells undergo a series of events to give rise to cartilage-ECM-producing chondroblasts and chondrocytes [21,22]. Chondrogenic differentiation involves gradual changes in the transcriptome, the proteome, and the electrome (i.e., the totality of all ionic currents) of the cells as they progress to more mature phenotypes [23]. Nearly all channels and transporters that mediate ionic currents across the plasma membrane of mature chondrocytes have been observed in CPCs as well, some of which exhibit interesting expression patterns and unique roles. There have been very few attempts to look at the global channelome expression in CPCs. This is not surprising, given the low abundance of these proteins to mediate ionic fluxes across the plasma membrane of CPCs, which are non-excitable cells. Using a combined approach of aminooxy biotinylation, glycocapture, and quantitative mass spectrometry, we described the surfaceome of migratory CPCs isolated from end-stage human knee OA cartilage [24]. We detected the protein level expression of K^+^ channel subunits (KCMA1, KCNJ2, KCNQ2); Ca^2+^ channel subunits (CA2D1, CA2D2, CA2D4, CAC1C, CAC1E, CAC1H); volume-regulated anion channel subunits (LRC8A, LRC8C, LRC8D); the mechanosensitive cation channel Piezo1; the TRPM2 and TRPM4 cation channels; and the VDAC1, VDAC2, and VDAC3 voltage-dependent anion-selective channels. We also carried out a transcript-level screening of ion channels and transporters in the same CPC cell line and established the composition of its channelome at the mRNA level [23]. Such global approaches are important for understanding the complexity of ionic currents in CPCs.

#### 2.1.1. Calcium Signalling Pathways

Calcium-dependent signalling pathways play a central role in regulating chondrogenic differentiation [25]. Ca^2+^ influx through ion channels in the plasma membrane (voltage-gated Ca^2+^ channels; TRPs; Orai; P2XRs; and other nonselective Ca^2+^ permeable channels) controls many physiological processes, including cell adhesion, cell growth, differentiation and proliferation, and cell death [26].

TRP vanilloid 4 (TRPV4) is a Ca^2+^-permeable nonselective cation channel that appears to be a key modulator in chondrogenic cells [27,28]. In mature chondrocytes, TRPV4 exhibits mechanosensory or osmosensory functions and controls responses to mechanical stimuli or osmotic stress, but it also has a critical role in skeletal development [29]. TRPV4 activation promotes chondrogenesis by inducing SOX9 transcription [27], and it is also involved in the chondrogenic differentiation of iPSCs [30]. TRPV4 channels also mediate the thermo-mechanotransduction process in human CPCs [31]. In addition to TRPs, the mechanically activated cation channel Piezo1 also regulates chondrocyte differentiation [32].

NMDA receptors expressed by mature healthy and OA chondrocytes are likely involved in mechanotransduction pathways [33,34] and/or in maintaining tissue homeostasis through regulating the chondrocyte circadian clock [35,36]. NMDAR expression and function are also required for chondrogenic differentiation [37].

Voltage-gated Ca^2+^ channels (VGCC) are expressed and are found functional in chondrocytes and chondrogenic cells alike [38,39]. VGCCs are critically important in the chondrogenic differentiation of micromass cultures [38]. The activity of the L-type Ca_V_1.2 is essential for chondrogenesis in situ in the developing limb [40]. Ca^2+^ signalling via verapamil-sensitive VGCCs plays a critical role in the chondrogenic response of MSCs exposed to hydrostatic pressure [41].

Extracellular nucleotides modulate cellular responses by activating ionotropic P2X or metabotropic P2Y purinergic receptors. Migratory CPCs have been reported to express various P2 receptor subtypes, which are required for an autocrine/paracrine purinergic mechanism that drives Ca^2+^ oscillations [42]. Primary limb bud-derived CPCs are also characterised by a wide array of P2 receptors; P2X4 was implicated to be a key mediator of chondrogenesis [43] and ATP oscillations essential for prechondrogenic condensation [42]. In contrast, P2Y1 has a chondro-inhibitory role in micromass cultures [44].

The internal Ca^2+^ stores of the endoplasmic reticulum in chondrogenic cells are also important for repetitive Ca^2+^ transients and cartilage formation. Replenishment of depleted stores (store-operated Ca^2+^ entry (SOCE)) is mediated by Orai channels in the plasma membrane [38]. SOCE is essential for the repetitive Ca^2+^ transients in CPCs [38,42].

#### 2.1.2. Potassium Channels

Potassium, the main intracellular ion, maintains the resting membrane potential (RMP), participates in the regulation of the cell volume, and links the physiological properties of the plasma membrane to the activity of intracellular metabolic pathways through intracellular-ligand (such as Ca^2+^ or ATP)-gated potassium channels [26]. Potassium currents mediated by potassium channels have been extensively documented in mature chondrocytes [19]. These include heteromultimeric voltage-gated potassium channels with a subunit composition of K_V_1.4 and K_V_1.6 [45,46], inwardly rectifying ATP-dependent potassium channels (K_ATP_) [47], as well as large and small calcium-activated potassium channels (SK and BK) [48,49].

However, much less is known regarding the potassium channelome of chondrogenic cells. In primary chondrifying micromass cell cultures, the protein level expression of K_V_1.1, K_V_1.3, and K_V_4.1 channel subunits has been documented, and the K^+^ channel blocker tetraethylammonium (TEA) reduced K_V_ currents, cell proliferation, cartilage-specific gene expression, and cartilage formation. Spontaneous Ca^2+^ transients were also suppressed by TEA [50], indicating a role of fluctuations of RMP in bringing about Ca^2+^ events [51]. Loss of the inwardly rectifying K^+^ channel Kir2.1 function impairs chondrogenic differentiation of iPSCs via the downregulation of master gene expression, by an impaired bone morphogenetic protein signalling pathway [52]. In another study, CPCs derived from normal and OA knee joint cartilage were shown to express BK, K_V_1.1, K_V_1.4, K_V_4.2, and EAG1 (*ether-à-go-go* or KCNH1) channels [53].

#### 2.1.3. Sodium Channels

While the transcript- or protein-level expression of voltage-gated sodium channels has been documented in both mature [54] and differentiating chondrocytes [50], the current understanding regarding their role in CPCs is generally lacking. This is not entirely surprising given that, at the rather depolarised RMP of developing and mature chondrocytes (in the realm of −20 to −40 mV), these channels would be permanently inactivated [19]. Epithelial sodium channels (ENaC) have been recognised in chondrocytes [19], but not yet in CPCs.

#### 2.1.4. Chloride Channels

The available literature on chloride channel expression and function in chondrocytes and CPCs is sparse [19]. Voltage-gated chloride channels (ClCs), besides being involved in the regulation of intracellular pH and cell volume homeostasis, are implicated to have a role in chondrogenesis as mutations of these genes result in abnormal skeletal development [55]. The transcript level expression of ClCs has been documented in chondrifying chicken mandibular mesenchymal cells, and important roles of ClCs in regulating chondrocyte proliferation and differentiation have been confirmed [55].

#### 2.1.5. Aquaporin Water Channels

Aquaporin water channels are expressed in chondrocytes, and they mediate the movement of water into and out of the cell during volume regulation [19]. Upregulation of AQP1 and AQP3 was observed during chondrogenic differentiation of MSCs, implicating a physiological adaptation to the mature chondrocyte phenotype [56].

### 2.2. Altered Ionic Homeostasis in Chondrocytes of OA-Afflicted Cartilage

Chondrocytes are routinely subjected to changes in water content of the articular cartilage, as the joint moves and is compressed over the course of a normal day. They therefore need robust mechanisms to cope with the changes in volume that occur due to these changes in extracellular osmolarity. Healthy chondrocytes are able to respond to these changes by reducing their volume and, thus, remain protected from mechanical stress [57]. As OA develops, the water content of cartilage increases with the early stages of the disease, increasing the volume of chondrocytes and potentially increasing the susceptibility of cells to damage [58,59]. This increase in water content corresponds to a deterioration of the collagen structure and proteoglycan loss from the cartilage, ultimately altering the ionic status of the tissue [60].

Many ion channels have been reported to mediate the chondrocyte cell response to the pro-inflammatory micro-environment in OA. For example, Piezo1, Piezo2, and TRPV4 are targets of potential mechanotherapeutic strategies to prevent OA-associated cartilage degeneration [61]. Altered NMDAR subunit expression is also involved in changing the chondrocyte phenotype in OA, at least partially via modulating the chondrocyte circadian clock [36]. The acid-sensing potassium channel (TASK-2), epithelial sodium channel (ENaC), Ca^2+^-activated chloride channel (anoctamin-1, TMEM16), Ca^2+^-activated potassium channels (K_Ca_3.1 and K_Ca_1.1), and aquaporin 1 (AQP1) are differentially expressed in OA vs. healthy chondrocytes [62]. When we specifically looked at the surfaceome of chondrocytes exposed to a pro-inflammatory environment, we found a differential expression of transporters including amino acid transporters (S38A2, S38A5), sodium/hydrogen exchanger 6 (SL9A6), and zinc transporter 1 (ZNT1) [63]. In line with altered ion channel expression and function, the global electrophysiological parameters (such as current density and cell capacitance) were different in chondrocytes isolated from OA knee joints compared to normal chondrocytes [53].

### 2.3. Ion Channels That Mediate Osteogenesis

There is ample literature available regarding ion channels with osteogenic roles. A recent review gave a highly detailed account of K^+^- and Ca^2+^-permeable channels in various osteogenic lineages and provided an excellent model explaining how their cooperation may drive forward the osteogenic differentiation process [64]. Herein, we list only the ones that have a particular importance and are in line with the latest trends.

Piezo mechanosensitive ion channels maintain muscle and bone mass, sense tendon stretch, and regulate senescence and apoptosis in response to mechanical stimuli. An excellent review article discussing Piezo functions in the musculoskeletal system has recently been published [65]. Along with Piezo1, TRPV4 also quite commonly arises in the context of Ca^2+^ response to mechanical stimulation. In MC3T3-E1 cells, both receptors appear to be functional in the osteoblastic mechano-transduction that results in reduced proliferation [66]. TRPM7 is also a common target for mechanical loading to stimulate bone formation, as results obtained in various models suggest [67,68,69]. TRPM7 is critically important in bone formation [70].

VDCCs are key regulators of intracellular Ca^2+^ homeostasis and control plasma membrane Ca^2+^ permeability in osteoblasts [71]. Both L-type Ca_V_1.2 and T-type Ca_V_3.2 appear to be involved in early long bone development in mice. The fact that osteocytes selectively lose L-type, and maintain T-type, VDCC expression suggests that the calcium homeostasis of post-proliferative cells of the osteoblast lineage is sustainable with low-voltage-activated channels [72].

Purinergic signalling is also a broadly accepted mediator of osteogenesis. The P2X4 receptor influences the differentiation of rat primary osteoblasts favouring the osteogenic lineage over the adipogenic lineage [73]. Similarly, P2X7 is expressed and functional in human primary osteoblasts; furthermore, NFATc1 was shown to be recruited to the P2X7 promoter to enhance its expression level in SaOS2 osteoblastic-like cells [74].

An extensive review summarised the recent literature on P2Y receptors in bone physiology and pathology [75]. Since P2Y1-2-4-6-11 are all Gq/11-coupled, activation of calcium signalling via the IP_3_R can be expected. These receptors are all expressed and play a significant role in various cell types of bone tissue. The available literature suggests that the outcome of intracellular signalling initiated via Gq/11 is strongly context dependent. P2Y1 receptor activity promotes osteoblast proliferation following low-intensity ultrasound stimulation [76], but in another setting, it also increases osteoclast formation and resorption [77]. P2Y2 and P2Y4 both play an inhibitory role in the differentiation and mineralisation of osteoblasts [78,79], while P2Y2 has also been shown to induce osteoclast activity [80]. P2Y6 enhances osteoclast activity [81] and survival [82], but also has an osteogenic effect in BM-MSCs of postmenopausal women [83]. While P2Y11 is not widely documented for its function in bone formation, it is a known inhibitor of breast cancer bone metastasis formation [84]. Other P2Y receptors (P2Y12-13-14) also have connections to osteoblast and/or osteoclast activity [85,86,87,88]; however, their intracellular signalling is not mediated by IP_3_R, at least not in a direct manner. IP_3_Rs are known players of osteoblast [89] and Gli1^+^ cell [90] intracellular Ca^2+^ homeostasis. There is also indirect evidence pointing at the importance of Ca^2+^ release from the ER store via IP_3_R during bone formation. TMEM38B encodes TRIC-B, a ubiquitous component of TRIC, a monovalent cation-specific channel involved in IP_3_R function. A deletion mutation in TMEM38B is associated with autosomal recessive osteogenesis imperfecta [91]. Correspondingly, a mouse model lacking TRIC-B displayed a highly similar phenotype [92]. The P2Y2 and P2X7 receptors are involved in mechano-transduction of osteoblastic C2C12-BMP cells [93]. Interestingly, the nucleotides that confer mechanical signals may be released via small repairable membrane injuries in bone-forming osteoblasts [94]. Another possible player in ATP release from osteoblasts is PANX3. It has been shown to determine the balance between proliferation and differentiation in several cell types of the musculoskeletal system [95]. In osteoblasts, PANX3 has been shown to facilitate ATP release, which can then act on P2 receptors to form an autocrine and paracrine loop [96]. The subsequent cytoplasmic Ca^2+^ increase drives calmodulin and Smad1/5 signalling and boosts osteogenic differentiation [96,97]. The P2Y and P2X receptors have also been implicated in the osteogenic differentiation of human MSCs. Adenosine may also promote osteogenesis via A2B receptors in the same cell type [98]. The P2X7 receptor is also known for its regulatory role in osteoclastogenesis. The suggested pathway includes Ca^2+^/calcineurin/NFATc1 [99].

Anion channels are also notable for their role in bone tissue formation and homeostasis. Anoctamin 1 is expressed in osteoclasts, and its expression demonstrates a positive correlation with osteoclast activity. Protein levels of this channel substantially increase in osteoporosis patients, demonstrating an interesting correlation and making it a highly interesting drug target candidate [100]. ClC-3 chloride channels are positive regulators of osteogenic differentiation in osteoblasts [101].

### 2.4. Ion Channels That Mediate Muscle Differentiation

During prenatal and postnatal development, committed myogenic precursors (myoblasts) align and fuse to generate multinucleated myofibres. Cells exit the cell cycle, and differentiation is induced, controlled by key transcription and specific regulatory factors. The high-level expression of myosin and actin proteins results in hypertrophy, ensuring the growth of myofibres. Myogenesis occurs during embryonic and postnatal growth phases and in adulthood during regeneration after injury [102].

The RMP in myoblasts becomes hyperpolarised as a prerequisite for myoblast fusion via the sequential expression of two different potassium channels. Initial events during myogenic differentiation are the activation of *ether-à-go-go* (EAG) K^+^ channels and Kir2.1 inward-rectifier K^+^ channels [103].

Myoblast fusion is strictly Ca^2+^-dependent. EAG and Kir2.1 have been shown to serve as a trigger for a small, but sustained Ca^2+^ influx through T-type VGCCs. These channels are expressed just before fusion and have intrinsic properties to produce a permanent Ca^2+^ current. EAG K^+^ channels are responsible for a rapid hyperpolarisation, which is followed by a further decrease in the RMP due to the opening of Kir2.1 channels. Besides the contributors of hyperpolarisation, several types of other ion channels, including ERG channels, SOCE channels, and volume-regulated anion channels (VRACs), proved to influence the RMP of fusion-competent myoblasts. The most-recent results suggest that the fusion-related hyperpolarisation is to set the RMP of myoblasts in a range that allows Ca^2+^ to enter through 1H T-type Ca^2+^ channels, causing a significant increase in the resting intracellular Ca^2+^ concentration, which activates numerous signalling cascades [103,104].

When TRPC1 was overexpressed in C2C12 myoblasts, the main parameters of SOCE significantly increased, while decreased STIM1 expression together with SERCA downregulation was observed. TRPC1 overexpression also resulted in morphological changes. Delayed differentiation and less-developed myotubes were observed, likely due to the decreased translocation of NFAT1 into the nucleus [105].

ATP-sensitive K^+^ channels (K_ATP_ channels) are nucleotide-gated ion channels that couple metabolism to excitability. They consist of Kir6.x pore-forming subunits encoded by KCNJ8 and KCNJ11 and regulatory SUR subunits encoded by ABCC8 and ABCC9. Zebrafish cardiac myocytes (CM) and vascular smooth muscle (VSM) cells express functional K_ATP_ channels with similar subunit composition, structure, and metabolic sensitivity to their mammalian counterparts [106]. KCNJ8/ABCC9 containing ATP-sensitive potassium channels are involved in VSM progenitor cell differentiation in the blood vessels of the brain, which fine-tune cerebral blood flow [107]. K_ATP_ channel modulators seem to be promising against certain types of muscle disorders such as hypokalemic periodic paralysis with vacuolar myopathies [108] or Cantú syndrome, a multiorgan condition caused by mutations in the ABCC9 and KCNJ8 genes, resulting in neuromuscular symptoms and skeletal malformations [109].

## 3. Calcium-Dependent Potassium Channels

The calcium-activated potassium (K_Ca_) ion channel family of proteins is closely related to the much larger 6 transmembrane (6TM) potassium channel superfamily that also includes the voltage-gated potassium channels [110]. The defining physiological property of the whole K_Ca_ family is that the channel open probability is increased by elevated intracellular calcium ions.

Within the K_Ca_ family, there are a number of distinct members, which can be separated on the basis of genetic identity, physiological properties, or pharmacological profile. Early work defined three types of K_Ca_ channels [111]: BK (Maxi K_Ca_ or the “Big”, large, or high-conductance K_Ca_, also known as K_Ca_1.1), SK (the small K_Ca_, also known as K_Ca_2.1/2/3 channels), and IK (an intermediate conductance K_Ca_, also known as K_Ca_3.1), although most studies were initially on BK. The distinction between BK and SK was non-trivial since the BK channel had a reported single-channel conductance of approximately 150 pS [112], whereas SK was a mere 14 pS [113]. Pharmacological separation of BK, SK, and IK is, and remains, partially successful; BK is sensitive to iberiotoxin and some other rather specific proteins, which spare both IK and SK. Newer synthetic chemicals distinguished SK and IK with only moderate selectivity. In genetic terms, now the gold standard in K_Ca_ research, the distinction is quite clear: BK is the product of the KCNMA1 gene; SK derives from KCNN1/2/3; IK is the product of the KCNN4 gene. Note there are also other closely related, but less well-studied members of this family (https://www.guidetopharmacology.org/GRAC/FamilyDisplayForward?familyId=69 accessed on 10 March 2023). 

Due to its high conductance, and thus amenability to patch-clamp electrophysiology, the K_Ca_1.1 channel is perhaps the best-studied. In terms of physiology, it is highly sensitive to voltage and Ca^2+^ [114]. For example, even in the virtual absence of Ca^2+^, it can still be driven to a high open probability state by sufficient depolarisation [114]. K_Ca_1.1 is sensitive to a range of pharmacological (relatively) selective inhibitors such as paxilline, charybdotoxin, and iberiotoxin and activators such as NS004 or NS1619 [115]. The primary K_Ca_1.1 gene (KCNMA1) is a true “*alpha*” subunit in the sense that it appears to form a functional channel on its own, but its properties will be modified in native tissue by the co-expression with one of a range of “*beta*” subunits such as KCNMB1. These accessory subunits can change voltage, Ca^2+^, and sometimes, pharmacological properties [116]. 

The structure of the KCNMA1 product (K_Ca_1.1) has been extensively studied; its 6TM structure forms a central conductive pore, and Ca^2+^ sensitivity is conveyed by the presence of an intracellular loop with a constitutive Ca^2+^ binding “bowl” [117], although SK can be activated with a Ca^2+^-/calmodulin-dependent mechanism [118,119].

All members of the K_Ca_ family are widely expressed in musculoskeletal [19,48,120] and other systems [121], including the brain [122]. Importantly, in the context of chondrocyte function, we found significant changes in both KCNMA1/B1 in OA [123]. In our own work with chicken CPCs, we found detectable expressions of KCNMA1, KCNMB1, KCNMB2, KCNMB4, KCNN1, KCNN2, and KCNN3 [124], and in a recent study of (mouse) adult skeletal muscle, for example, we detected all these, plus KCNMB43 and KCNN4 [125].

The functional role of K_Ca_ is a fascinating question and still requires some speculation; firstly, K_Ca_1.1 has such a high conductance that any activity from this channel is likely to directly impact the RMP of a cell, which can, in turn, be important for a wide range of cellular functions [126]. Secondly, there has been the suggestion that K_Ca_1.1 may play a role in apoptosis, which is critical to development [127]. Thirdly, we ourselves showed that K_Ca_1.1 was involved in cell volume regulation in adult chondrocytes [59,128], supporting the earlier proposal of an “osmolyte channel” in these cells. Finally, and more generally, K_Ca_ seems to have a role in coupling Ca^2+^ entry to RMP [129]. Intuitively, this idea seemed unlikely, because the global Ca^2+^ concentration may not rise sufficiently during normal physiology to open the K_Ca_. However, we and others have shown by experiment and calculation that, whilst global [Ca^2+^] may not rise sufficiently to activate K_Ca_, local [Ca^2+^] can rise sufficiently in an appropriate microdomain [120,129,130]. This may be a widespread biological principle and would suggest that the evolutionary advantage of the broad range of subtly different K_Ca_ channels is to allow a high degree of flexibility in potential Ca^2+^ entry—RMP coupling mechanisms.

### 3.1. Calcium-Dependent Potassium Channels in MSCs, CPCs, and Chondrocytes: Expression and Function

#### 3.1.1. K_Ca_ Channels in MSCs

The expression of all the main subtypes of K_Ca_ channels was detectable in MSCs with different origins of species and tissues. BK channel expression was observed in mouse bone-marrow-derived (BM) MSCs [131], in rat BM- and adipose-derived (AD) MSCs [132,133], in human BM- and AD-MSCs, and iPSCs [134,135,136]. IK channels were detected in mouse BM-MSCs [131], in rat MSCs [132], in dog BM-MSCs [137], and in human BM- and AD-MSCs [134,138]. However, SK channels were mainly detected in human AD- and endometrium-derived MSCs [134,139].

Various roles were associated with ionic currents mediated by K_Ca_ channels in MSCs. Gene silencing of BK channels significantly downregulated PPARγ and osteocalcin (biomarkers of adipogenesis and osteogenesis, respectively) in human MSCs [135]. Knock-down of BK channels downregulated osteocalcin, osteopontin, and RUNX2 in a rat MSC-derived osteoblast cell line [140]. Blocking BK channel activity with paxilline or shRNA significantly inhibited proliferation in human MSCs [135]. Intriguingly, paxilline had an opposite effect on iPSC-derived human MSCs [136]. Although the presence of BK channels in the plasma membrane of endometrium-derived MSCs was reported to be cell-cycle-dependent, with significantly lower levels in the G2M phase, the BK channel inhibitors iberiotoxin or charybdotoxin did not influence cell cycle progression [141]. In contrast, the role of the intermediate conductance IK channels in MSC proliferation seems to be less controversial. Blocking IK channels either with a pharmacological approach (using CLT) or siRNA decreased cell cycle progression in rat BM-MSCs [132]. Attenuated IK channel function by TRAM-34 or siRNA strongly suppressed mechanical stretch-induced proliferation of rat BM-MSCs [142]. Similar results were observed in murine MSCs, where the inhibition or knock-down of IK channels resulted in the downregulation of cyclin D1 and E, attenuating proliferation [143]. A synergy between BK channel function and platelet lysate has been shown to improve the migration of rat AD-MSCs [133].

As an attempt to better understand how external mechanical cues are coupled with calcium signalling in stem cells, a study was conducted on human-endometrium-derived MSCs [139]. Using a combination of cell-attached and inside-out patch-clamp, highly localised Ca^2+^ entry via stretch-activated channels was observed, which triggered BK channel activity. BK channels were clustered together with stretch-activated channels in functional mechanosensitive domains in the plasma membrane of the stem cells.

#### 3.1.2. K_Ca_ Channels in CPCs

In a transcriptome-based screening, we observed the mRNA expression profiles for BK, IK, and SK channel subunits in CPCs [23]. According to the Affymetrix microarray transcriptome analysis, KCNMA1 (BK) and KCNN4 (IK) were upregulated in CPCs compared to MSCs, whereas KCNN3 (SK) was unchanged. Interestingly, the beta subunit of BK channels (KCNMB4) was downregulated significantly in CPCs. A higher level of KCNMA1 protein expression in CPCs was confirmed by quantitative mass spectrometry, western blotting, and immunocytochemistry [24]. In the presence of selective BK channel inhibitors (100 nM IBTX and 1 μM paxilline), the CPC proliferation rate was significantly reduced, but had no significant effect in MSCs. Paxilline treatment upregulated osteogenic marker gene expression, while the chondrogenic marker SOX9 remained unchanged. Paxilline treatment significantly increased the fibronectin-guided and random migration parameters of CPCs. These data indicated that BK channels may play an inhibitory role in migratory CPCs [23].

#### 3.1.3. K_Ca_ Channels in Chondrocytes

Ca^2+^-dependent K^+^ channels have been widely documented in chondrocytes [19]. BK and IK channel expression was observed in murine [45], canine [144], and human chondrocytes [145]. The K^+^ currents mediated by these channels not only modulate, but also dominate the chondrocyte RMP [51]. A primary effect is hyperpolarisation of RMP, and a secondary consequence is voltage-independent Ca^2+^ transport via TRP and CRAC channels.

The BK and SK channels are involved in mechano-transduction, cell volume regulation, chondrogenesis, and apoptosis [146]. SK channels were activated after cyclic pressure-induced strain in human articular chondrocytes [147]. Cyclic stretching significantly increased the expression of BK channels in equine articular chondrocytes [148]. BK channels have a central role in the membrane-stretch-activated current in equine articular chondrocytes [48]. BK channel response to external mechanical stimuli was significantly reduced in chondrocytes isolated from OA patients [149], and they have a differential expression in pathological cartilage [59].

C-type natriuretic peptide (CNP) stimulates growth plate chondrocytes through natriuretic peptide receptor 2 (NPR2), which involves cGMP-dependent protein kinase (PKG). PKG stimulates BK channels, leading to hyperpolarisation of the membrane, which facilitates TRPM7-mediated Ca^2+^ influx and increases the phosphorylation of Ca^2+^-/calmodulin-dependent protein kinase II (CaMKII) [150]. In turn, Ca^2+^-/calmodulin-dependent signalling enhances ACAN mRNA expression under compression forces in bovine articular chondrocytes [151].

BK channels are regulated not only by intracellular Ca^2+^ or the RMP, but also by reactive oxygen species (ROSs) [152]. In the context of chondrocytes, BK channel activation by ROSs is an exciting, novel approach to better understand the pathomechanism of OA, as an age-related imbalance in ROSs’ production is involved in the disease [153].

There is evidence that changing the osmolarity of the chondrocyte microenvironment alters BK channel activity. Both hypotonic and hypertonic solutions activate BK channels [154], resulting in K^+^ efflux and membrane hyperpolarisation in human chondrocytes. While a hypo-osmotic condition upregulated BK channel expression in equine articular chondrocytes, hyperosmotic stress had the opposite effect [148]. The effects of osmotic changes were modulated by the ERK1/2 and p38 MAPK pathways.

### 3.2. Calcium-Dependent Potassium Channels in CPCs and Osteocytes: Expression and Function

BK channels are now widely accepted as a necessary element for bone formation and osteoblast function. There is already a body of evidence available from in vivo and in vitro studies that underlines the importance of these channels. Expression data from human osteoblasts are available from as early as 2008 [155], but recent years have seen a definite shift up in this area. 

BK knockout mice are characterised by impaired bone formation and osteoblast activity. Notably, BK appears to regulate bone formation at least partly via the integrin pathway; the binding of the BK α subunit with the integrin β1 protein has been confirmed in osteoblasts, and FAK-ERK1/2 signalling is also shown to be involved in conveying the effects of BK knockout [156]. The connection between BK channels and FAK has aroused interest in BK channels as a possible trigger for the signalling cascade induced by mechanical strains in osteoblasts [157]. Rat ROS17/2.8 osteoblast cells following BK knockout demonstrated a significantly decreased ability for proliferation and mineralisation [140]. BK channel modulators also affect the number of MG63 cells [158]. Evidence from KCNMA1 knockout mice suggests the Wnt/β-catenin signalling pathway to be among the main mechanisms of how BK channels influence bone homeostasis [159]. Er-xian Decoction (EXD) is a well-known prescription drug widely used to prevent and treat climacteric syndrome and osteoporosis in China. It is now suggested that its osteogenesis-promoting effects are attributable to upregulating BKα expression [160].

BK channels are also actively researched in vascular smooth muscle cells (VSMCs), whereas the BK channel opener NS1619 significantly alleviated vascular calcification and decreased the mRNA expression of the osteogenic genes OCN and OPN, as well as RUNX2 [161]. The lncRNA “Antisense RNA 1 of KCNMA1” (KCNMA1-AS1) appears to be an important regulator in bone homeostasis. This RNA gene has an elevated expression level in osteoporosis patient samples, while its silencing in human BM-MSCs resulted in elevated osteogenic differentiation via the cochlin-miR-1303 axis [162].

The functional expression of IK channels was first demonstrated in human osteoblasts by Hirukawa et al. in 2008 [155]. These findings suggest that IK channels regulate osteoblast RMP, which is an important contributor to their physiological Ca^2+^ homeostasis. IK channels also positively regulate the cell proliferation of MC3T3-E1 mouse pre-osteoblasts via intracellular Ca^2+^ signalling, which is an important early step of osteogenesis [163]. The osteogenic role of IK channels is notable enough, and this has also recently emerged as a possible target to prevent vascular calcification [164]. The specific K_Ca_3.1 inhibitor TRAM-34 turned out to be an excellent target in a model of vascular smooth muscle cell calcification as it interfered with the calcification-related signalling of NF-κB and TGF-β and reduced the mRNA expressions of osterix, osteocalcin, and matrix-metalloproteinases (MMPs)-2/-9 [164]. On the other hand, receptor activator of NF-κB-ligand (RANKL)-activated murine bone marrow macrophage (BMM) cultures demonstrated a marked upregulation of K_Ca_3.1 during osteoclastogenesis. Treatment with TRAM-34 significantly reduced the expression of osteoclast-specific genes alongside decreased osteoclast formation under both inflammatory and non-inflammatory conditions [165]. There is emerging evidence that IK channels may participate in conveying the effects of mechanical stimuli. In rat bone-marrow-derived MSCs, the stretch-induced increase in cell proliferation and cell cycle progression largely depends on K_Ca_3.1 function as proven by pharmacological or genetic inhibition experiments [142].

Data regarding the role of SK channels in bone-related cell types is very scarce. In 2000, Gu et al. measured three different types of K^+^ currents in MLO-Y4 (murine-osteocyte-like) cells. In a relatively low percentage (22%) of cells, a slowly activating, voltage-activated TEA-insensitive current was observed, which was strongly inhibited by apamin, a selective inhibitor of SK channels [166]. In human articular chondrocytes, SK channels have been shown to participate in the mechanotransduction pathway, which also involves α5β1-integrin, the actin cytoskeleton, and tyrosine protein kinases [147]. A similar role could be anticipated in bone tissue, where mechanical forces also play a significant role in the homeostasis and bone mass maintenance [167]. Additionally, the K_Ca_2.1 SK channel is discussed in the context of Ewing sarcoma (EwS), a rare and highly malignant bone tumour occurring mainly in childhood and adolescence [168]. The same team has unveiled that the expression of this channel is directly regulated by EWSR1-FL1, which is the disease-driving oncoprotein. Interestingly, K_Ca_2.1 appears non-conductive in EwS cells by patch-clamp electrophysiology, meaning that KCNN1 mRNA could instead be applied as a prognostic marker in EwS and also meaning that these cells might be more vulnerable to hypo-osmotic stress [168].

### 3.3. Calcium-Dependent Potassium Channels in Muscle Progenitor Cells and Myogenesis: Expression and Function

The activation of potassium channels and the resulting membrane hyperpolarisation in myoblasts is essential for myogenesis. BK channels are widely expressed in skeletal muscle, myocardium, and smooth muscle. In KCNMA1 knockout (BK^−/−^) rats, decreased muscle fibre area and lower grip strength were detected. Furthermore, BK^−/−^ rats showed muscle atrophy, decreased muscle strength, slowness, and tremor compared to wild-type animals [169]. Cardiac muscle physiology was also adversely affected: cardiac systolic/diastolic function and heart rate were reduced in BK^−/−^ rats, and the wall of the left ventricle became significantly thinner.

BK channels are also involved in TGF-β1-induced differentiation of human AD-MSCs into vascular smooth muscle cells [170]. TGF-β1-induced differentiation of hASCs significantly increased the expression of the α-subunit of the BK channel, but not the SK and IK channels.

IK channels are also expressed in skeletal myoblasts and are involved in the regulation of muscle differentiation [171,172]. Administration of the Ca^2+^-activated K^+^ channel opener DCEBIO facilitates muscle differentiation and causes hypertrophy in differentiating C2C12 myoblast cells. These effects were mediated by mitochondrial IK channels, and the increased phosphorylation level of Akt, a known skeletal muscle hypertrophy factor [173].

### 3.4. Calcium-Dependent Potassium Channels in Cells of the Synovium: Expression and Function

In synovial tissues, the role of BK channels has primarily been investigated in the context of RA. K_Ca_1.1 channels have recently emerged as a broadly investigated therapeutic target for RA [174]. Besides immune cells, resident FLSs play a pivotal role in RA pathogenesis [175]. FLSs are non-immune cells and are considered key effector cells of RA by producing cytokines that maintain inflammation and various proteases that contribute to the destruction of articular cartilage [176].

There is accumulating evidence from various RA models that many of the pathological functions of FLSs can be attributed to BK channels. Tanner et al. [177] showed that the interactions between FLSs and effector memory T (T_EM_) cells in rats with collagen-induced arthritis are regulated by K_Ca_1.1 and K_V_1.3. K_Ca_1.1 is required for FLSs to stimulate T_EM_ cell proliferation and migration. Conversely, K_V_1.3 on T_EM_ cells contributes to enhancing K_Ca_1.1 expression on FLSs. Haidar et al. [120] documented that the increased BK activity in FLSs in an inflammatory arthritis model correlates with FLS activation following pro-inflammatory cytokine (TNFα and IL1β) treatment. KCNMA1 mRNA was significantly upregulated, while BK channel β-subunit expression levels were also altered (KCNMB1/2 were downregulated, whereas KCNMB3 was upregulated). There is a correlation between K_Ca_1.1 channel density and FLS invasiveness [178]. Treatment with a BK channel blocker when the clinical signs of arthritis started to manifest inhibited the progression of the disease in a rat model of pristane-induced arthritis (PIA); joint and bone damage were reduced, and the invasiveness and proliferation of FLS were constrained [179]. Disturbances in calcium homeostasis, the sustained phosphorylation of Akt, and talin recruitment to β1-integrins appear to be key signalling events, by which K_Ca_1.1 regulates β1-integrin function and, therefore, the invasiveness of FLSs [180]. BK channels have also been identified in human RA patient samples as the major potassium channel expressed at the plasma membrane of FLSs [181]. In the same human RA samples, blockade of K_Ca_1.1 in FLS disrupts the calcium homeostasis and inhibits a number of clinically unfavourable traits (proliferation, production of VEGF, IL-8, and pro-MMP-2 and migration and invasion) of these cells [181].

After studying synovial fluid mesenchymal progenitor cells from normal and early OA samples, Bertram et al. concluded that a number of ion channels had an altered expression at the mRNA level. Interestingly, KCNMA1 expression remained unchanged [182]. Remarkably, the differences among the above findings may indicate that a strong inflammatory context might be obligatory for the synovial expression of BK channels to change.

## 4. BK Channels in Calcium Oscillations in CPCs

Both MSC and CPC cells exhibit Ca^2+^ oscillations, which are periodic elevations of the intracellular Ca^2+^ concentration ([Ca^2+^]_i_) [183,184,185,186]. Early steps of chondrogenic differentiation are driven by these repetitive intracellular Ca^2+^ signals, which rely on Ca^2+^ influx from the extracellular environment [183]. Thus, it is reasonable to assume that Ca^2+^ oscillations are associated with a similar temporal fluctuation of the RMP. In this study, we propose a model in which the intrinsic pacemaker function of chondrogenic cells is based on a membrane clock mechanism operated by a coordinated interplay between various voltage-gated ion channels of the plasma membrane (Figure 1).

This model involves BK channels, L-type, voltage-gated Ca^2+^ channels (Ca_v_1.2), the non-voltage dependent epithelial Na^+^ channel, and the Na^+^/Ca^2+^ exchanger (NCX), which have all been shown to be essential for Ca^2+^ oscillations [23,39,40,146,184,186,187]. In addition, the role of the “leak” K^+^ channel TASK-2 and some voltage-gated delayed rectifier K^+^ channels (K_V_1.1, 4.1, and 1.3) is also considered [4,50,145]. The baseline MP of chondrogenic cells is quite depolarised (~−21 mV), suggesting that the TASK-2 background K^+^ conductance and the delayed rectifier channel conductance are balanced by the constitutively active Na^+^ conductance of the ENaC. At this membrane potential, the Ca_v_1.2 channels are open, which also contributes to the depolarised MP and mediates Ca^2+^ influx. As the Ca_v_ and BK channels are often clustered together, the local [Ca^2+^] in the proximity of BK channels reaches values high enough to open them [188,189,190,191]. BK represents the predominant K^+^ conductance in chondrogenic cells and is known to link Ca^2+^ signalling to MP [23,192]. In doing so, the BK current will hyperpolarise the MP, which in turn deactivates the Ca^2+^ current. Elevated [Ca^2+^]_i_ is also the key to reverse the MP from its most-hyperpolarised point, as it enhances NCX activity, which creates a depolarising current owing to its 3Na^+^/1Ca^2+^ stoichiometry. While the NCX (in cooperation with the plasma membrane Ca^2+^ pump (PMCA)) restores the [Ca^2+^]_I_, the BK channel deactivates, and the inward cationic currents will prevail. This will cause the MP to spontaneously depolarise back to baseline, which will initiate the next membrane clock cycle again by activating the Ca^2+^ current. Pharmacological inhibition of any component of this chain clamps the MP in a hyperpolarised or depolarised state and interrupts the cycle [23,40,184]. Additional inward Ca^2+^ currents, mediated by ionotropic purinergic receptors or transient receptor potential channels of the vanilloid family (TRPV4), may also regulate the [Ca^2+^]_i_ depending on the stage of development [30,43,183,193].

## 5. K_Ca_ Channels in Musculoskeletal Pathologies

BK channels have been well documented to play a pivotal role in regulating several physiological processes including musculoskeletal homeostasis, neuronal excitability, and pain signalling. Thus, their therapeutic potential may be of particular interest for clinical practice [174,194].

Channel disorders, also known as channelopathies, are implicated in various human diseases related to striated and smooth muscle. In skeletal muscle, BK channels with different functional and pharmacological properties have been described [195], and they are involved in a great variety of functions. Collected data support that slow-twitch rat fibres display a rather increased BK expression associated with low Ca^2+^ sensitivity and insensitivity to channel activator acetazolamide, whereas drug-sensitive BK channels of fast-twitch rat fibres with high Ca^2+^ permeability show lower activity [195,196]. It has been proposed that ageing potentiates BK channel currents of fast-twitch fibres, promoting abnormal fast–slow fibre remodelling, which occurs with muscle disuse [197]. BK channels appear to actively participate in fibre transition in a cell-line-based and in a rat ischemia–reperfusion model of hyperkalaemia [198,199]. In hypokalaemia periodic paralysis (hypoPP) patients, BK channel expression in the membrane fraction of muscle cells was detected in significantly lower amounts compared to the cytosolic compartment, emphasising the role of channel redistribution in pathological states [200]. BK channel activators (e.g., acetazolamide, dichlorphenamide) administered together with Na^+^ channel inhibitors may alleviate the weakness and myotonia in patients [201]. BK channel agonists can also appropriately depolarise skeletal muscle fibres in paralysis and hypoPP [202,203]. In agreement with this, recent findings obtained using BK^−/−^ rats demonstrated reduced skeletal muscle fibre area, speed, and movement compared with wild-type littermates. The cardiac muscle of BK^−/−^ rats also displayed the shrinkage of the muscle fibre area, lower cardiac systolic/diastolic function, and heart rate [169].

BK channels may also be one of the key determinants of vascular tone regulation [201], since polymorphisms of α and β1 channel subunits are correlated with the prevalence of asthma and hypertension. Gain-of-function polymorphism within KCNMB1 encoding the β1-subunit may lead to severe diastolic hypertension [204] and alteration of baroreflex function [205]. Specific Glu65Lys polymorphism of β1 may decrease systolic and diastolic blood pressure [206]. Mutations of the α-subunit are also associated with increased risk of hypertension and myocardial infarction [207].

BK channels have been identified as the major potassium channels in FLSs derived from patients with RA. Unsurprisingly, blockade of the BK channel with paxilline disrupts Ca^2+^ homeostasis and prevents RA-FLS proliferation and invasion by inhibiting cytokine and MMP production. Therefore, it provides a potential target candidate for RA therapy [181].

The often debilitating chronic pain prevailing in degenerative joint diseases such as RA and OA is the result of autoimmune and local inflammatory processes, in which both peripheral and central sensitisation may be implicated [208]. Accumulating data suggest the role of BK channels in pain modulation [194,209,210]. BK channels may contribute to the inhibition of neurotransmitter release by preventing overexcitation, since iberiotoxin was shown to enhance the activation of spinal glycinergic neurons [211]. Peripheral nerve injury robustly alters the plasticity of both primary afferents and spinal dorsal horn neurons, resulting in central sensitisation and, eventually, neuropathic pain. BK channel expression was substantially decreased following L5-L6 spinal nerve ligation, and pharmacological blockade with iberiotoxin significantly reduced the mechanical paw withdrawal threshold of animals [212].

Intriguingly, BK channel activation also principally negatively affects pain signalling at supraspinal levels such as the anterior cingulate cortex, the motivational–affective centre and a key hub for anxiety and pain perception, although overexpression of the synaptic and non-synaptic BK channel β4 subunit evoked mechanical allodynia along with anxiety-like behaviour [213].

## 6. Conclusions and Perspectives

K_Ca_ channels, especially BK channels, play fundamental roles in cells of the differentiating and mature tissues of the musculoskeletal system, including cartilage, bone, synovium, tendon, and muscle. Unsurprisingly, these ion channels are increasingly recognised to be involved in pathophysiological conditions including pathologies of the central nervous system (seizures, epilepsy, autism, and intellectual disability, cerebral ischemia and hypoxia, hypertension, obesity, and diabetes mellitus) [214]. The versatility of BK channels is also attributable to the fact that the β-subunits, which are characterised by a specific tissue distribution, modify channel kinetics, pharmacological properties, and the apparent Ca^2+^ sensitivity of the α-subunit. Additionally, alternative splicing isoforms confer different tissue expression and function to the channel [201], representing one of the main mechanisms of BK channel diversity. In fact, the splicing variants may help the design of molecules selectively targeting the subtypes expressed in different tissues of the musculoskeletal system. The use of drugs selectively targeting tissue-specific BK channel isoforms is a promising strategy in the treatment of various disorders, including hyperkalaemia and hypokalaemia periodic paralysis in the case of the muscular system [201]. Unfortunately, the tissue- and cell-specific distribution of BK channel subunit isoforms has not been mapped in detail in relation to the progenitor cells of the musculoskeletal system. Therefore, this is an important area to be addressed by future research.

BK channel subunits are also subject to posttranslational modifications including protein phosphorylation, lipidation, glycosylation, and ubiquitination [215]. These posttranslational modifications control the number, properties, and regulation of BK channels in specific cell types. Posttranslational regulation is important for the temporal control of channel function from the millisecond to hours timescale. However, the spatial localisation is also important for BK channel function. Although posttranslational modifications are important determinants of the temporal or spatial control of BK channels [215], we know relatively little about the spatiotemporal dynamics of BK channels in cells of the musculoskeletal system. This subject also warrants further studies.

Another important point regarding K_Ca_ channels in various settings is that they are unlikely to act independently. The way K_Ca_ channel subunits link to intracellular proteins, metabolites, and pathways, as well as their modulation by extracellular natural agents, is highly relevant in many physiological processes [216]. A high number of proteins are known to interact with BK channels (Figure 2). The interactants may be involved in trafficking/targeting of BK channels to specific plasma membrane compartments (such as actin (ACTA2), caveolins (CAV1, CAV2, CAV3), cereblon (CRBN), catenin (CTNNAL1), or microtubule-associated protein 1A (MAP1A)); removal or degradation of BK channels such as cullin-1 (CUL1) or F-box only protein (FBXO7); and modulation of BK channel properties, including regulators subunits such as KCNMB1, B2, B3, B4, LRRC26, as well as cAMP-dependent protein kinase subunit alpha (PRKACA). Given that BK channels are linked to many diseases, including musculoskeletal disorders, understanding cell-specific BK channel interactomes will help us to expand our knowledge on how altered BK channel functions contribute to these conditions. This knowledge could be then exploited to design pharmaceutical therapies that block or activate specific regulatory mechanisms by targeting the interaction between BK channels and their interacting partners.

## Figures and Tables

**Figure 1 ijms-24-06796-f001:**
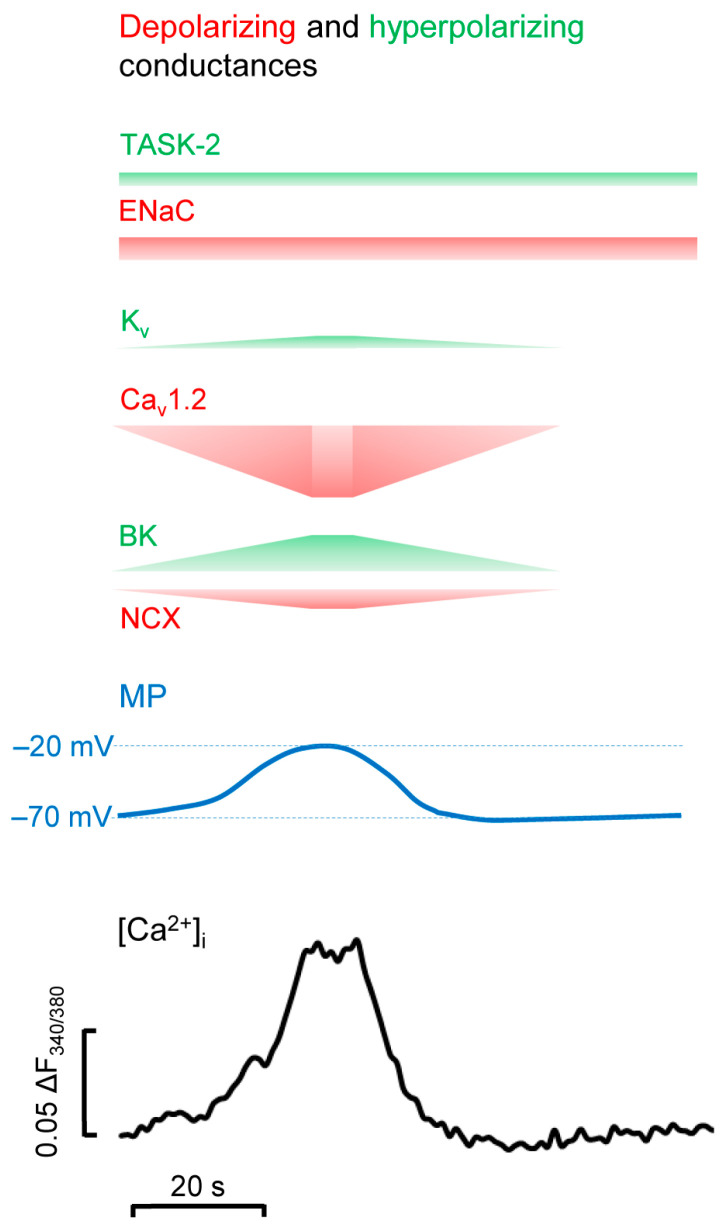
A model describing ion conductance fluctuations accounting for the oscillation of the membrane potential and intracellular Ca^2+^ concentration in chondrogenic cells. An original record of a transient elevation of the intracellular Ca^2+^ concentration ([Ca^2+^]) in a single chondrogenic cell (as reported previously in [42]) is shown in black. Simultaneous temporal change of the membrane potential (MP) and the relative magnitude of depolarising and hyperpolarising ionic conductances were estimated and are illustrated in blue, red, and green, respectively.

**Figure 2 ijms-24-06796-f002:**
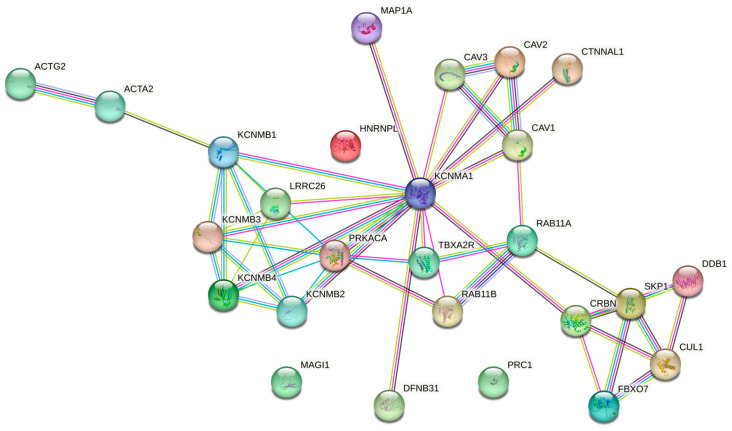
The STRING interaction network for KCNMA1, showing the top 25 interactants. Edge colours are as follows: known interactions: light blue, from curated databases; magenta, experimentally determined; predicted interactions: green, gene neighbourhood; dark blue, gene co-occurrence; others: lime, text-mining; black, co-expression; purple, protein homology.

## Data Availability

Not applicable.

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
