# Peer review of "Ca^2+^-Activated K^+^ Channels in Progenitor Cells of Musculoskeletal Tissues: A Narrative Review"

_ijms, 2023, doi:10.3390/ijms24076796_

Round 1
Reviewer 1 Report
The authors of this manuscript discuss the role of ion channels in the proliferation and differentiation and migration of pro-musculoskeletal parent cells.
The aim of the authors is to highlight the knowledge concerning KCa channels in musculoskeletal progenitor cells that could be exploited for the development of new therapeutic strategies for musculoskeletal disorders and to set up stem cell-based therapies. Autoi highlight the need to focus on underlying biological mechanisms.
For this the authors discuss the role of different ion channels encoded by the following genes: KCNJ2, KCNQ2, TRPV4, TRPM2, TRPM4 at high channels, paying particular attention to the BK encoded channels of the KCMA1 gene (alpha subunit) where they show their fundamental role in cells of the differentiating and mature tissues of the musculoskeletal system, including cartilage, bone, synovium, tendon, and muscle. Since their role is evident in many diseases, including musculoskeletal disorders, the authors have made a synthesis to highlight scientific evidence for understanding channel interactomes.
1) I would recommend a thorough grammar check of the entire manuscript and a weighted edition of the introduction if too much information is presented non-fluently.
2) Adviser to add data concerning the ATP-sensitive K + channel including the KCNJ8, KCNJ11, ABBC8 and ABCC9 genes, as the KATP channels have together with the BK channels a sunergic role in the regulation of proliferation, differentiation and above all the KATP channels have an impact on muscle development and maintenance, I recommend expanding the bibliography with works that deal with the topic, for example:
1- F. Maqoud, R. Scala, M. Hoxha, B. Zappacosta, and D. Tricarico, “ATP-sensitive potassium channel subunits in the neuroinflammation: novel drug targets in neurodegenerative disorders.,” CNS Neurol. disorder. Drug Targets, Jan. 2021, doi: 10.2174/1871527320666210119095626
2- https://doi.org/10.3390/cells12060928
3-https://doi.org/10.1016/j.maturitas.2014.12.003
Author Response
Our team of authors thanks the journal editor and the reviewers for their time and thorough review of our work. We appreciate their questions, comments, and critiques. We have endeavoured to revise the manuscript and provide a response that addresses these questions/concerns in a balanced manner. We have provided a detailed point-by-point response to the reviewers and the editor below. In this response, we provide the original reviewer comments in black and our comment-by-comment response in blue. In the revised manuscript, all changes to the text are indicated by highlights in the document revised-manuscript-with-tracked-changes.docx.
We hope that the reviewers and the editor find the revised manuscript improved and suitable for publication in the International Journal of Molecular Sciences.
Kind Regards,
Csaba Matta, PhD
corresponding author
Reviewer #1
The authors of this manuscript discuss the role of ion channels in the proliferation and differentiation and migration of pro-musculoskeletal parent cells.
The aim of the authors is to highlight the knowledge concerning KCa channels in musculoskeletal progenitor cells that could be exploited for the development of new therapeutic strategies for musculoskeletal disorders and to set up stem cell-based therapies. Authors highlight the need to focus on underlying biological mechanisms.
For this the authors discuss the role of different ion channels encoded by the following genes: KCNJ2, KCNQ2, TRPV4, TRPM2, TRPM4 at high channels, paying particular attention to the BK encoded channels of the KCMA1 gene (alpha subunit) where they show their fundamental role in cells of the differentiating and mature tissues of the musculoskeletal system, including cartilage, bone, synovium, tendon, and muscle. Since their role is evident in many diseases, including musculoskeletal disorders, the authors have made a synthesis to highlight scientific evidence for understanding channel interactomes.
Author response: We thank the reviewer for their considered review of our manuscript, and for their kind words regarding its presentation. Please find our detailed response to the reviewer comments below.
1) I would recommend a thorough grammar check of the entire manuscript and a weighted edition of the introduction if too much information is presented non-fluently.
Author response: Thank you for this suggestion. We have carried out a thorough English grammar and vocabulary check on the manuscript text.
2) Adviser to add data concerning the ATP-sensitive K + channel including the KCNJ8, KCNJ11, ABBC8 and ABCC9 genes, as the KATP channels have together with the BK channels a sunergic role in the regulation of proliferation, differentiation and above all the KATP channels have an impact on muscle development and maintenance, I recommend expanding the bibliography with works that deal with the topic, for example:
1- F. Maqoud, R. Scala, M. Hoxha, B. Zappacosta, and D. Tricarico, “ATP-sensitive potassium channel subunits in the neuroinflammation: novel drug targets in neurodegenerative disorders.,” CNS Neurol. disorder. Drug Targets, Jan. 2021, doi: 10.2174/1871527320666210119095626
2- https://doi.org/10.3390/cells12060928
3-https://doi.org/10.1016/j.maturitas.2014.12.003
Author response: Thank you for this comment. We agree with the reviewer that the ATP-sensitive K+ channel coded by the KCNJ8, KCNJ11, ABBC8 and ABCC9 genes have not been discussed in the original version of the manuscript. We have now endeavoured to discuss these KATP ion channels as well, especially in the context of muscle development and maintenance. We have also added new references as suggested. Please see the updated text on page 8, lines 387–398 in the revised version of the manuscript.
Reviewer 2 Report
The submitted narrative review article describes Ca2+ activated K+ channels in progenitor cells of musculoskeletal (MSK) tissues. The introduction provides a clear and concise overview of the MSK system and its progenitor cells, as well as the need for a deeper understanding of these cells to develop effective treatment options for MSK disorders such as osteoarthritis (OA). In the main body, the authors provide detailed information about the ion channels involved in progenitor cell differentiation, focusing on the calcium-dependent potassium (KCa) channels. They discuss the three types of KCa channels - BK, SK, and IK - and their role in different MSK progenitor cells. The review is well-organized into coherent subsections and sources are appropriately cited. In the conclusion, the authors emphasize the fundamental role of KCa channels, especially BK channels, in the MSK system, and suggest that understanding cell-specific BK channel interactomes could lead to the development of pharmaceutical therapies that modulate or target these channels. Overall, the article is unbiased, well-written, and provides valuable insights into the role of KCa channels in MSK tissues.
Minor comments:
1. Line 29: In the abstract, the authors mention BK channels without clarifying that they are a type of KCa ion channel. This may confuse readers who are not familiar with the topic. The authors should add more details about BK channels in the abstract to provide better context.
2. Page 7: The role of P2X7 and P2X4 receptors is nicely explained. However, the authors should provide more elaboration on the function of P2Y receptors in osteogenesis.
Author Response
Our team of authors thanks the journal editor and the reviewers for their time and thorough review of our work. We appreciate their questions, comments, and critiques. We have endeavoured to revise the manuscript and provide a response that addresses these questions/concerns in a balanced manner. We have provided a detailed point-by-point response to the reviewers and the editor below. In this response, we provide the original reviewer comments in black and our comment-by-comment response in blue. In the revised manuscript, all changes to the text are indicated by highlights in the document revised-manuscript-with-tracked-changes.docx.
We hope that the reviewers and the editor find the revised manuscript improved and suitable for publication in theInternational Journal of Molecular Sciences.
Kind Regards,
Csaba Matta, PhD
corresponding author
Reviewer #2
The submitted narrative review article describes Ca2+ activated K+ channels in progenitor cells of musculoskeletal (MSK) tissues. The introduction provides a clear and concise overview of the MSK system and its progenitor cells, as well as the need for a deeper understanding of these cells to develop effective treatment options for MSK disorders such as osteoarthritis (OA). In the main body, the authors provide detailed information about the ion channels involved in progenitor cell differentiation, focusing on the calcium-dependent potassium (KCa) channels. They discuss the three types of KCa channels - BK, SK, and IK - and their role in different MSK progenitor cells. The review is well-organized into coherent subsections and sources are appropriately cited. In the conclusion, the authors emphasize the fundamental role of KCa channels, especially BK channels, in the MSK system, and suggest that understanding cell-specific BK channel interactomes could lead to the development of pharmaceutical therapies that modulate or target these channels. Overall, the article is unbiased, well-written, and provides valuable insights into the role of KCa channels in MSK tissues.
Author response: We thank Referee 2 for their considered review of our manuscript, and for their kind words regarding its presentation. We accept their comments and critique regarding the weaknesses of the original version of the manuscript. Please find our detailed response to the reviewer comments below.
Minor comments:
Line 29: In the abstract, the authors mention BK channels without clarifying that they are a type of KCa ion channel. This may confuse readers who are not familiar with the topic. The authors should add more details about BK channels in the abstract to provide better context.
Author response: Thank you for raising this point. We agree with the reviewer that the original version of the abstract can be a bit confusing to the readers who are not specialists in this field. We have therefore added the following sentence to the abstract: “This review article focuses on the big conductance (BK) KCa channels.”
Page 7: The role of P2X7 and P2X4 receptors is nicely explained. However, the authors should provide more elaboration on the function of P2Y receptors in osteogenesis.
Author response: Thank you for this comment. We agree with the reviewer that the original version of the manuscript could benefit from a more elaborate discussion on the function of P2Y receptors in osteogenesis, despite this topic not being in the focus of this narrative review, given that P2Y receptors are metabotropic purinergic receptors, rather than ligand gated ion channels. As suggested by the reviewer, we have added a more detailed discussion on the role of P2Y receptors in osteogenesis on page 7, lines 322–333 in the revised version of the manuscript.